# Exploring the Dynamics of Shikimate Kinase through Molecular Mechanics

Pedro Ojeda-May

High Performance Computing Center North (HPC2N), Umeå University, S-90187 Umea, Sweden; pedro.ojeda-may@umu.se

**Abstract:** Shikimate kinase (SK) enzyme is a suitable target for antimicrobial drugs as it is present in pathogenic microorganisms and absent in mammals. A complete understanding of the functioning of this enzyme can unveil novel methods to inactivate it. To do this, a clear understanding of SK performance is needed. Previously, the chemical step of SK was studied in detail, but a study of longer-term scale simulations is still missing. In the present work, we performed molecular dynamics (MD) simulations in the μs time scale that allowed us to explore further regions of the SK energy landscape than previously. Simulations were conducted on the wild-type (WT) enzyme and the R116A and R116K mutants. We analyzed the dynamics of the enzymes through standard MD tools, and we found that the global motions in the mutants were perturbed. These motions can be linked to the observed undetectable binding affinity of the WT enzyme and the R116A and R116K mutants.

**Keywords:** shikimate; kinase; mutants; molecular; dynamics; binding

## 1. Introduction

Several primitive organisms, including pathogenic bacteria, rely on the Shikimate pathway to synthesise essential compounds for their functioning, for instance, vitamins [1–3]. Because this pathway is absent in host mammals, in particular humans, one can potentially design antimicrobial drugs that target any of the steps involved in the pathway to inactivate the microorganism without affecting the hosts.

Shikimate kinase (SK) is the enzyme involved in the fifth step along the Shikimate pathway and its function is to catalyse the conversion of ATP and shikimic acid (SKM) to ADP and shikimate-3-phosphate (S3P) through the conveyance of the ATP terminal phosphoryl group. As SK is essential in the pathway, a good understanding of its performance is important to understand the possible ways to inhibit its functioning which can in turn lead to a blocked pathway.

SK is composed of three main domains shown in Figure 1: a flexible LID which has an opening and closing feature also observed in other protein kinases [4,5], a binding domain SB where the binding to the SK substrate occurs, and a stiffer CORE domain which is composed of α-β-α motives. The binding pocket of SK contains several conserved residues D33, R57, R116, and R132 which have been found critical for catalysis as mutations at the level of these amino acids-produced inactive enzymes [4].

A detailed description of the energy landscape (EL) of SK will reveal its performance at the different stages of the catalysis [6]. Up to now, only the EL region corresponding to the chemical step of SK has been studied in detail with high-level quantum chemical (QC) methods [7,8]. Other EL regions farther away from the chemical step such as the region corresponding to the free apo protein, have not been explored currently.

Extensive sampling of these regions is crucial for understanding the dynamics of the enzyme and in particular the changes in dynamic motions when mutations are induced. Some of these SK motions, specifically the LID opening/closing motion, have been suggested to play a dynamic role in the binding of substrates and catalysis [9]. In fact, LID

motions are rate-limiting in other kinases such as adenylate kinase [5]. Due to the time scales involved, currently, only molecular dynamics methods (MD) and enhancements [10] can resolve these motions.

In the present work, we conducted MD simulations in the μs time scale for three variants of the SK enzyme: the wild-type (WT) enzyme, and two mutants of the conserved amino acid R116, i.e., R116A and R116K. These two mutants are relevant as they displayed a dramatic reduction in the activity of the enzyme [4]. They will be our computational lab to study a potential link between the dynamics and the reduction in the binding activity of the WT enzyme.

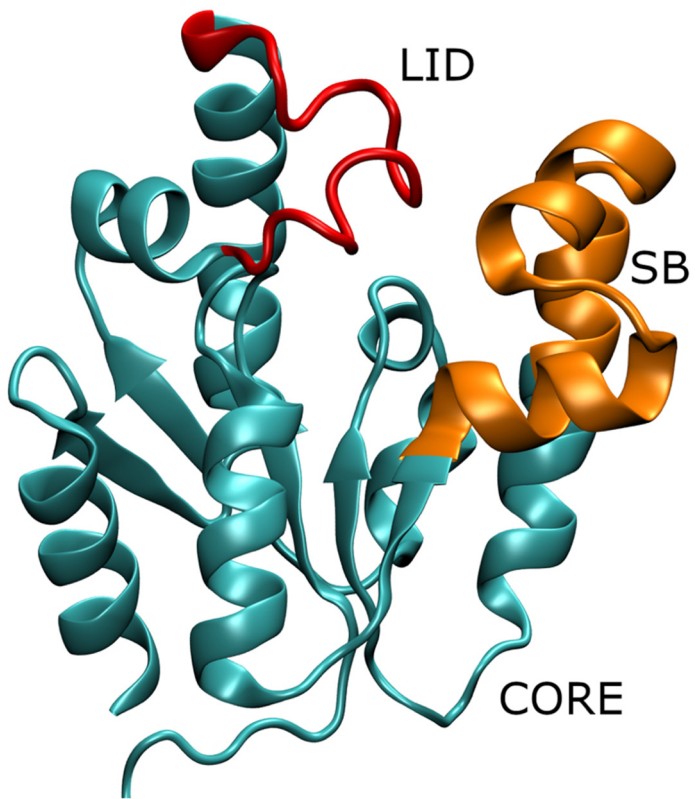

**Figure 1.** The three main domains in the Shikimate Kinase (SK) enzyme structure: the binding SB (in orange), the LID (in red), and the CORE domains (in turquoise).

## 2. Materials and Methods

### 2.1. Simulation Setup

The initial structure of *Helicobacter pylori* SK was retrieved from the Protein Data Bank (PDB ID: 3MUF) [4]. The CHARMM-GUI [11] server was used to obtain the initial input files for all simulations. Three different variants for the SK enzyme were considered: one for the wild-type (WT) enzyme, and the other two for the mutants R116A and R116K, respectively. The CHARMM-36 force field [12–14] was used to model the atom types and interactions. The missing tail residues were included and modelled according to the CHARMM-GUI tools.

The initial protein structures were solvated with TIP3P waters [15] in cubic periodic boxes whose dimension was computed by adding a 20 Å solvation layer to the largest centered atomic position. NaCl salt was added to the system with a 150 mM concentration. The resulting initial structures for the three variants contained 68,100, 70,689, and 66,177 atoms for the WT, R116A, and R116K cases, respectively.

The GROMACS simulation package (v. 2021) [16–22] was employed to solve Newton's equations and to obtain the dynamic trajectories. Three independent repetitions for each variant were performed. Electrostatic interactions were handled with the particle mesh

Ewald (PME) method [23,24] by employing the fourth order of cubic interpolation method and a grid size of 1.2 Å. The short-range interactions for the van der Waals and electrostatic terms were truncated at 12 Å. During the simulation, all hydrogen bonds remained constrained with the LINCS algorithm [25,26].

The systems were first energy-minimised, with a maximum allowed value for the forces of 239 kcal/mol nm, using harmonic restraints on the backbone and side chain atoms with force constants of 95.6 kcal/mol nm and 9.5 kcal/mol nm, respectively. After the initial minimisation, the systems were equilibrated during 5 ns using a time step of 1 fs and keeping the same restraints as in the minimisation process. This step was done within the NVT ensemble obtained through the Nosé-Hoover thermostat with a coupling constant of 1 ps.

A further equilibration step in the NPT ensemble was conducted during 50 ns, using a 2 fs time step where all restraints were removed. The barostat employed was the Parrinello-Rahman method [27] with the following parameter values: coupling constant and compressibility of 5 ps and $4.51 \times 10^{-5} \, \text{bar}^{-1}$, respectively, and a reference pressure of 1 bar. During this equilibration step, the temperature was controlled with the Nosé-Hoover thermostat with the same coupling constant value as previously. The equilibration step was followed by data production during 1 μs for each variant and with three independent repetitions resulting in a total of 3 μs per variant.

### 2.2. Analysis

Standard methods for the analysis of MD trajectories were performed on the collected data, i.e., the root mean square deviation (RMSD), the root mean square fluctuation (RMSF), and the principal component analysis [28] (PCA). In order to cover a large conformational space in the analysis of our results, the three independent trajectories were considered for each variant. For all calculations the C, O, N, and $C^{\alpha}$ atoms were selected.

VMD [29] built-in tools were employed to compute the RMSD and RMSF values. The PCA was conducted with the GROMACS built-in tools.

## 3. Results and Discussion

### 3.1. Root Mean Square Fluctuation (RMSF)

In Figure 2b we can observe the RMSF for the three variants. Starting with the WT case, we noticed that the enzyme shows its largest RMSF fluctuations in the region that contains the LID domain, supporting the idea previously proposed, based on the analysis of B-factors of crystal structures [30], that this region is highly flexible. Interestingly, LID motions have been suggested to play a critical role in binding and catalysis mainly because conserved residues, responsible for the proper positioning of substrates, are located in this region [9]. For instance, Arg 116 which is conserved among the SK proteins of other organisms, as revealed by sequence alignment [4,30], belongs to the LID domain, see Figure 2a.

The region corresponding to the SB domain also shows large fluctuations revealing that both the SB and LID domains can undergo large motions in the absence of substrates. These motions were reported previously in SK holo structures [9]. Interestingly, these motions are also present in other kinase proteins [5]. In the same Figure, we observe that most of the backbone atoms in the CORE domain display the lowest fluctuations except for the atoms in β region, more specifically the $3_{10}$ helix atoms which show fluctuations as high as those in the SB domain.

The fact that some regions in the protein structure display high action when others stay nearly static shows that energy is partitioned in the enzyme and supports the idea of a choreography of motions [6] for SK. The location of the regions in the protein with the highest fluctuations can be seen in Figure 2a.

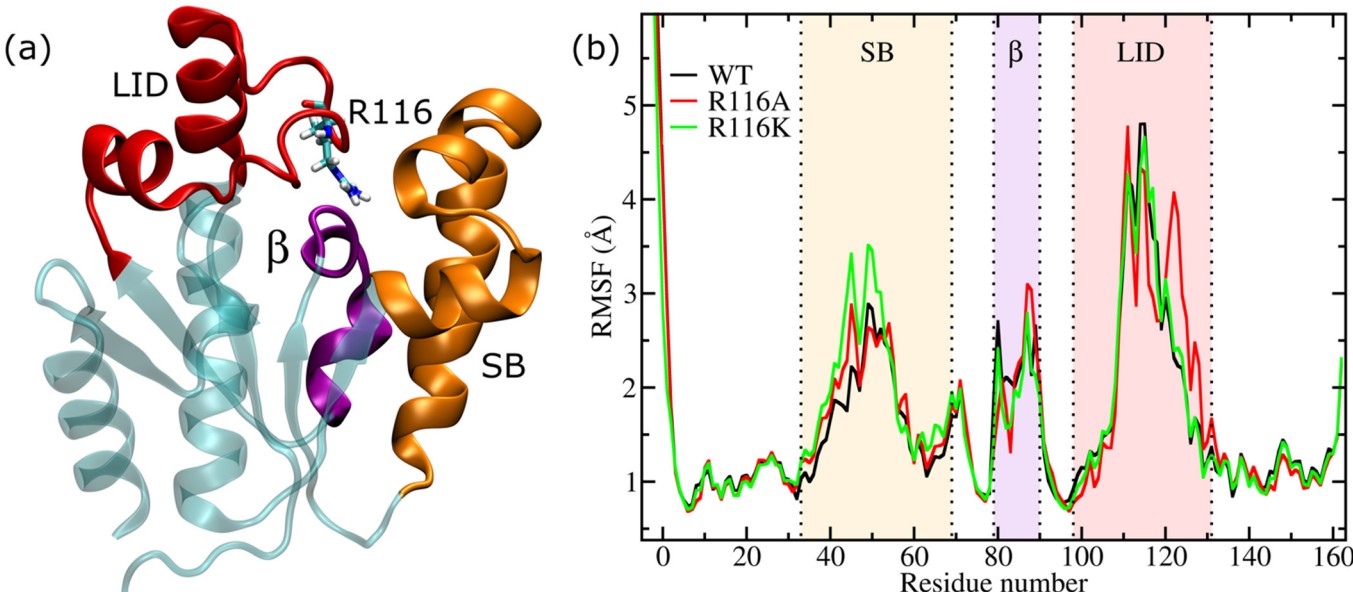

**Figure 2.** RMSFs results for the WT and the mutants R116: (**a**) regions with the largest fluctuations, i.e., the LID region (in red), the SB region (in orange), and the $3_{10}$ helix which is part of the β region in the CORE domain (in purple); (**b**) RMSF values for the three variants and the most affected regions marked with similar colours as in (**a**) for the regions.

The conserved Arg 116 among SKs is essential for catalysis as mutations of this residue destroy the activity of the enzyme [4]. The undetectable binding activity of the mutants R116A and R116K [4] can be explained by the modifications of the local electrostatic environment [31] of the guanidino group. For instance, after substrates' binding, the local modifications can cause that the proper positioning of the reactants is not optimal for catalysis leading to a decreased activity. However, before the binding occurs, the change in the global motions could lead to different protein-substrates interactions, for instance through different hydrogen bond patterns. The latter can also contribute to reducing the enzyme's binding activity [31–36]. It is interesting to notice that an increased level of motions of relevant binding domains has been found in other protein kinases, such as Adenylate kinase, upon mutation [37].

To quantify the impact of mutations on the global motions of the SK enzyme, we also performed MD simulations for the R116A and R116K mutants. The quantification was achieved through the RMSF calculations, and the results are presented in Figure 2a,b. We observed that these two mutants also display large fluctuations in similar regions as the WT variant, but the magnitude of the fluctuations is different, for instance, R116A shows larger fluctuations than the other two variants in the LID region. It also shows fluctuations similar in magnitude to the WT variant in the SB region.

In the case of R116K, the RMSF does not deviate considerably from the WT fluctuations in the LID region, but larger fluctuations are observed in the SB region than in the other two variants. We also noticed that both mutants exhibit deviations from the WT RMSFs in the β region. The fact that R116A exhibits strong fluctuations in the LID and R116K keeps the fluctuations stable (with reference to the WT variant) suggests that the charge and length of the Arg residue are essential for keeping the fluctuations in the same magnitude as in the WT case because these two features are similar in the Lys substituent. Regarding the SB region, both mutants perturbed the motions of the WT enzyme.

Overall, the motions of the three domains of SK are perturbed in the mutants. This perturbation can contribute to the drastically decreased binding affinity of the mutants as the substrate in the SB region can spend more time adopting the proper pose for binding. Perturbation of motions, specially those of binding domains, by mutants has also been observed in other protein kinases [37] which suggests that this can be a common feature

in other enzymes that exhibit similar domain motions. In this way, mutants destroy the choreography of enzymes exercised during the long protein evolution process [6].

### 3.2. Principal Component Analysis (PCA)

To analyse the extent of global dynamics of the enzymes we performed a principal component analysis (PCA) for all variants and projected the trajectories onto the first two principal components. The PCA revealed that the size of the configurational space sampled by WT was similar to that of R116A, see Figure 3. This analysis also showed that R116K performed the most extensive exploration of the configurational space.

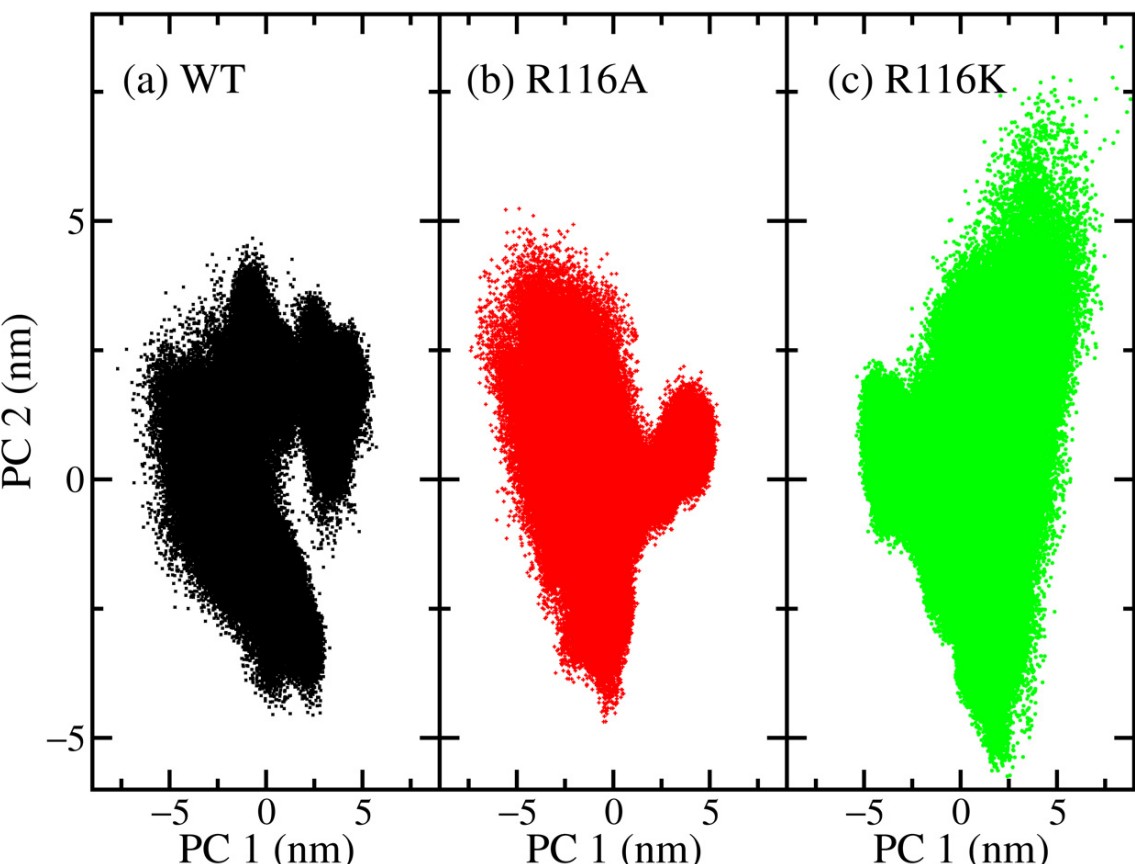

**Figure 3.** Projection of the MD trajectories for the WT, R116A, and R116K onto the two first principal components: (**a**) WT, (**b**) R116A, and (**c**) R116K variants. WT and R116A sample a configurational space of similar size and R116K explores a broader space.

A concerted motion of the LID and SB domains upon binding was suggested previously based on the analysis of crystal holo structures [9]. In our case, the LID and SB domain motion was observed in the first principal component projection for the WT, see Figure 4, with a similar feature for the other variants. The concerted feature was observed through visual inspection of the projected trajectory. This unveils that the concerted motion is already present in the apo structure.

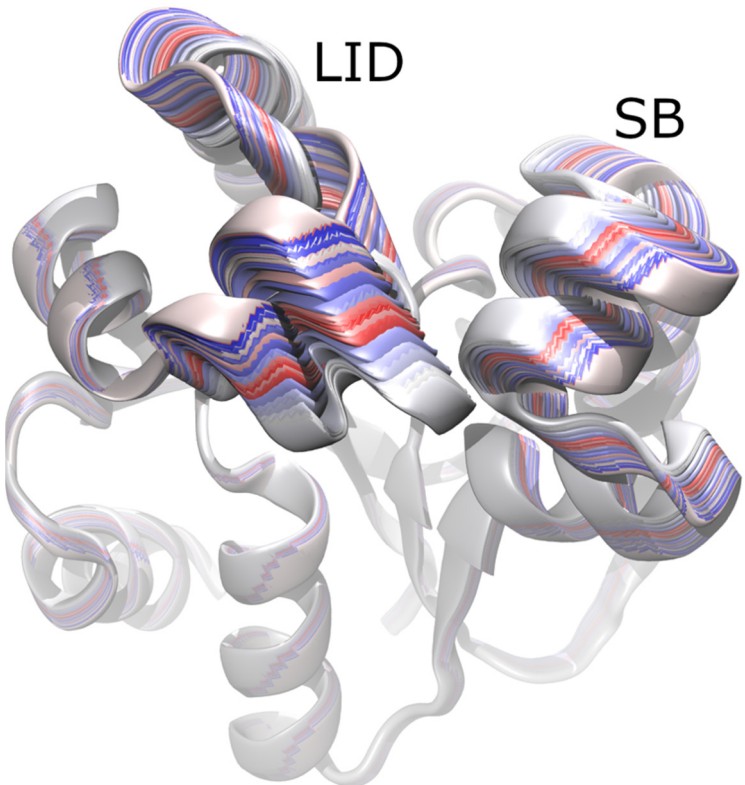

**Figure 4.** Trajectory projected on the first principal component showing the opening and closing motion of the LID and SB domains.

*3.3. Root Mean Square Deviation (RMSD)*

We were also interested in monitoring the changes in the populations of closed and open conformations for the variants considered in the present study. To do this the RMSD was computed for all systems with reference to the crystal structure which is in a closed state. Then, histograms with the number of counts for each state, encoded by RMSD bins, were prepared. In this way, configurations with high RMSD can be linked to more open structures and vice versa, these histograms can be seen in Figure 5.

In Figure 5a we noticed that the WT enzyme exhibits only one peak around 2.0 Å and it was able to sample the (1.3 Å, 4.6 Å) range. Notice that the distribution is skewed in the direction of more "open" conformations which are necessary for allowing the substrates to enter the binding pocket.

Regarding the mutants, we observed that in R116A the peak is shifted toward more open conformations (2.2 Å) and the covered range is slightly broader (0.9 Å, 4.6 Å) than in the WT case. In addition to this, a second small peak appears which shows a more "closed" character. In the case of R116K, two peaks are observed at 1.7 Å and 2.6 Å, respectively. This mutant exhibited the largest covered configurational space with a range of RMSD values of (1.2 Å, 5.8 Å).

The different distributions for the populations observed in the mutants with reference to the WT enzyme can contribute to the non-detectable binding activity of this enzyme as the population is shifted towards more closed states and this can in turn hinder the proper binding of the substrate. Notice that this contribution can be on top of the local electrostatic contribution that modifies the protein-substrates interactions after substrates enter the binding pocket.

Previously, it was suggested that arginines in the binding pocket of adenylate kinase control the closed-to-open conformational dynamics [37]. This suggestion is verified in our simulations for SK because a single arginine is able to induce changes in the conformational motions.

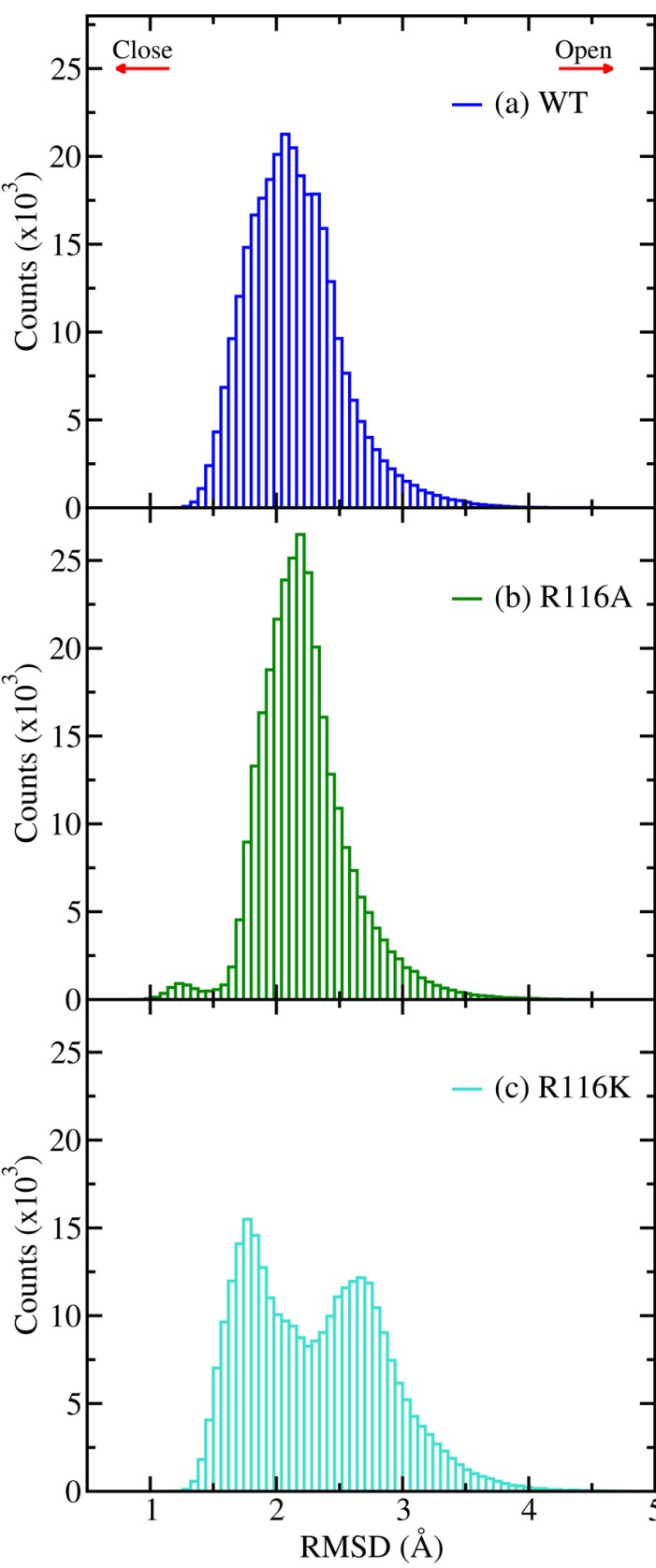

**Figure 5.** Distributions of "closed" and "open" conformations for (**a**) the WT, (**b**) R116A, and (**c**) R116K variants. Notice that mutants cover a broader interval than the WT and populations are shifted towards more "closed" states. The direction of "closed" and "open" conformations is indicated by the arrows.

## 4. Conclusions

In the present work, we studied the dynamics for the SK enzyme in the μs time scale. Three variants of this enzyme were considered, the WT case and the R116A, and R116K mutants. The analysis of the dynamics of the WT enzyme, through RMSF calculations, showed that the LID and SB domains and a small region of the CORE domain ($3_{10}$ helix) displayed the largest fluctuations and the other regions remained almost stiff. The PCA supported the experimental observation of the concerted motion of the LID and SB domains. RMSD calculations allowed us to visualise the population ensemble which is unimodal for the WT variant and skewed towards more open states. These results from the WT enzyme provided us with a baseline for the analysis of the dynamics.

The two mutants displayed perturbed motions with reference to the baseline, for instance, here larger RMSF were observed in the LID and SB domains. Interestingly, perturbation of motions by mutants have been reported previously for other protein kinases suggesting that this can be a common feature in other enzymes. Also, the covered configurational space was broader in the mutants as revealed by the PC and RMSD distribution analyses.

Overall, we found that the population ensembles for the mutants were altered with reference to the WT population specially in the shift towards more "closed" states. These altered populations can hinder the substrates to enter the binding pocket and can explain the decreased binding activity of the enzyme. In this way, the mutants destroy the choreography of the SK achieved during protein evolution.

**Funding:** This research received no external funding.

**Data Availability Statement:** Not applicable.

**Acknowledgments:** This research was conducted using the resources of High Performance Computing Center North (HPC2N) at the Swedish National Infrastructure for Computing (SNIC).

**Conflicts of Interest:** The authors declare no conflict of interest.

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
