# Peer review of "Exploring the Dynamics of Shikimate Kinase through Molecular Mechanics"

_biophysica, doi:10.3390/biophysica2030020_

Round 1

Reviewer 1 Report

The author has decided to investigate a particular enzyme (Shihikimate kinase) that appears in pathogenic microorganisms and not in humans. As such, understanding this enzyme better could provide the basis for designing small molecule drugs that aim at blocking the function of this enzyme. 

The author has used standard MD all-atom simulations. Here, the choice of the force-field and the time length of the trajectories are crucial. The authors has made adequate choices and the analysis is relevant based on standard software and tools. Hence, the results seem to be reliable and simulations have been carried out carefully. The lack of experimental confirmation is of course a minus, but as a simulation study, I believe that the presentation is complete.

The manuscript is well-written, literature appropriate and results well-presented and discussed with appropriate figures. One thing that could improve is of course analysis of more properties (e.g. free energy), but this is not a crucial thing.

Overall, the study is interesting and, since data are reliable, I am happy to recommend publication of this work as is. 

Author Response

We thank the Reviewer for the suggestions about possible ways to improve our work. Regarding the comments of the Reviewer:

Reviewer: The lack of experimental confirmation is of course a minus, but as a simulation study, I believe that the presentation is complete.

In contrast to well-studied enzymes such as Adenylate kinase, Shikimate kinase has not been extensively studied. The present work can motivate experimental groups to explore this enzyme with techniques such as NMR to monitor global motions. In its present form, our results are predictions of possible future experiments.

Reviewer: The manuscript is well-written, literature appropriate and results well-presented and discussed with appropriate figures. One thing that could improve is of course analysis of more properties (e.g. free energy), but this is not a crucial thing.

This is an exciting topic for future studies and we will consider it with high priority.

Thanks,

The Authors.

Reviewer 2 Report

Manuscript entitled “Exploring the dynamics of Shikimate Kinase through molecular mechanics” is an article focused on molecular dynamic simulation (MD) of Shikimate Kinase (SK) enzyme in micro-second scale and provided insights into lower affinity of mutant SK with respect to WT. Authors have chosen most conserved residue R116A/K since it showed dramatically reduced activity to compared with WT to test the effect of mutation on substrate affinity. Authors performed RMSF, RMSD and PCA analysis to elucidate the effect of mutation at the structural level. Through their analysis authors have nicely showed that that mutant SK exists in ensemble of different conformations and lot of flexibility is encoded in mutant compared to WT which explains why mutant SK lower affinity towards substrate compared to WT. Authors have done a good job of summarizing results from analysis and nicely presented the data.

General comments:

1.     Abstract, introduction and results were nicely written, and I don’t see much improvement in the sections. Also methods were adequately written.

2.     I would suggest authors to show the conservation of residue R116 in species through multiple sequence alignment, makes readers to follow the easily and make clear idea about the mutation.

3.      Authors presented the results from their analysis clearly, but I feel that they have done poor job of discussing them and I would suggest authors to discuss the results elaborately and compare their results with existed literature in similar family of kinases having similar fold. so that the study can be broad scope and imply towards understanding similar protein fold containing kinases.

4.     With above suggestions- I would recommend the manuscript to accept in the journal.

Author Response

We thank Reviewer 2 for the insightful comments that we have addressed in this version of the manuscript. Here there is a reply to these comments:

Reviewer: I would suggest authors to show the conservation of residue R116 in species through multiple sequence alignment, makes readers to follow the easily and make clear idea about the mutation.

Sequence alignment was previously published [Journal of Bacteriology, 23, 8156 (2005), PLoS One. 2012; 7(3): e33481]. Thus, instead of doing this alignment again, we refer the reader to the cited work. The phrase was added:

“For instance, Arg 116 which is conserved among SK proteins of other organisms, as revealed by sequence alignment [4,30],…” (lines 116-117)

Reviewer: Authors presented the results from their analysis clearly, but I feel that they have done poor job of discussing them and I would suggest authors to discuss the results elaborately and compare their results with existed literature in similar family of kinases having similar fold. so that the study can be broad scope and imply towards understanding similar protein fold containing kinases.

Thanks for pointing us to the discussion of the results and the comparison to existing literature. The comparison of our results for SK with other protein kinases was done through Adenylate kinase which has similar dynamical motions as SK. Several phrases/paragraphs were added:

  1. “Interestingly, these motions are also present in other kinase proteins” (lines 121-122)
  2. “It is interesting to notice that an increased level of motions of relevant binding domains has been found in other protein kinases, such as Adenylate kinase, upon mutation [37].” (lines 140-142)
  3. The Ref. 37 was added in this review.
  4. “Previously, it was suggested that Arginines in the binding pocket of Adenylate kinase control the closed-to-open conformational dynamics [37]. This suggestion is verified in our simulations for SK because a single arginine is able to induce changes in the conformational motions.” (lines 232-234)

The scope of the results was enlarged through the following paragraph:

“Perturbation of motions, specially those of binding domains, by mutants has also been observed in other protein kinases [37] which suggests that this can be a common feature in other enzymes.” (lines 170-173)

And the phrase:

“Interestingly, perturbation of motions by mutants have been reported previously for other protein kinases suggesting that this can be a common feature in other enzymes.” (lines 255-256)

We hope that the Reviewer is satisfied with our changes,

The Authors.